# Arbovirus Transmission in Australia from 2002 to 2017

**DOI:** 10.3390/biology13070524

**Published:** 2024-07-15

**Authors:** Elvina Viennet, Francesca D. Frentiu, Emilie McKenna, Flavia Torres Vasconcelos, Robert L. P. Flower, Helen M. Faddy

**Affiliations:** 1Research and Development, Strategy and Growth, Australian Red Cross Lifeblood, Kelvin Grove, QLD 4059, Australia; emiliemckenna1996@gmail.com (E.M.); flavia.torres@research.usc.edu.au (F.T.V.); rflower@redcrossblood.org.au (R.L.P.F.); hfaddy@usc.edu.au (H.M.F.); 2School of Biomedical Sciences, Centre for Immunology and Infection Control, Queensland University of Technology, Brisbane, QLD 4001, Australia; francesca.frentiu@qut.edu.au; 3School of Health, University of the Sunshine Coast, Petrie, QLD 4052, Australia

**Keywords:** arbovirus, weather patterns, blood donation, cluster analysis, Australia

## Abstract

**Simple Summary:**

Arboviruses, such as Ross River virus, Barmah Forest virus, and dengue virus, have the potential to pose a public health threat in Australia. Certain arboviruses pose significant concerns due to their potential impact on the safety and supply of blood transfusion products, necessitating a greater in-depth understanding of spatiotemporal patterns. This study analyzed the spatiotemporal distribution of eight arboviruses of public health significance in Australia from 2002 to 2017, using Geographic Information System mapping and space–time scan statistics. Key weather variables, including rainfall, temperature, and humidity, were found to influence arbovirus incidence rates and blood donation counts. The findings highlight regions at risk and offer insights for improving public health interventions.

**Abstract:**

Arboviruses pose a significant global public health threat, with Ross River virus (RRV), Barmah Forest virus (BFV), and dengue virus (DENV) being among the most common and clinically significant in Australia. Some arboviruses, including those prevalent in Australia, have been reported to cause transfusion-transmitted infections. This study examined the spatiotemporal variation of these arboviruses and their potential impact on blood donation numbers across Australia. Using data from the Australian Department of Health on eight arboviruses from 2002 to 2017, we retrospectively assessed the distribution and clustering of incidence rates in space and time using Geographic Information System mapping and space–time scan statistics. Regression models were used to investigate how weather variables, their lag months, space, and time affect case and blood donation counts. The predictors’ importance varied with the spatial scale of analysis. Key predictors were average rainfall, minimum temperature, daily temperature variation, and relative humidity. Blood donation number was significantly associated with the incidence rate of all viruses and its interaction with local transmission of DENV, overall. This study, the first to cover eight clinically relevant arboviruses at a fine geographical level in Australia, identifies regions at risk for transmission and provides valuable insights for public health intervention.

## 1. Introduction

Over the past five decades, several arboviral diseases have escalated into a public health emergency, rapidly emerging and/or re-emerging in various countries [1]. Recent examples include dengue virus (DENV) [2], chikungunya virus (CHIKV) [3,4], West Nile virus (WNV) [5], and Zika virus (ZIKV) [6]. DENV and CHIKV affect many tropical and sub-tropical regions, and can cause severe symptoms, such as hemorrhagic fever, death, and long-lasting arthralgia. In 2023 alone, over six million DENV cases and more than 6000 related deaths were reported from 92 countries/territories [7]. Meanwhile, CHIKV caused approximately 500,000 cases and over 400 deaths globally in the same year [8]. WNV outbreaks have significantly impacted health in the United States of America since 1999 [9,10] and more recently in Europe [5], while ZIKV, known for causing neonatal malformations, was declared a Public Health Emergency of International Concern in 2016 and has been recorded in 89 countries and territories.

Australia hosts more than 75 known arboviruses [11], with DENV, Barmah Forest virus (BFV), Ross River virus (RRV), Murray Valley encephalitis virus (MVEV), West Nile virus Kunjin strain (WNV_KUN_), and Japanese encephalitis virus (JEV) being the most clinically relevant for humans. RRV, the most notified Australian arbovirus, has also been identified in several Pacific Islands, including Fiji [12]. While BFV occurs less frequently than RRV, it is among the most common arboviruses to affect the Australasian region [13]. Both arboviruses are endemic to Australia [14,15] and exhibit similar clinical presentations (joint pain, fatigue, headache, and fever) [16], and share common vector species, such as *Aedes vigilax*, *Aedes procax*, and *Culex annulirostris* [15,17,18,19]. Local DENV transmission occurs in the warmer months in northern Queensland, Australia, and is initiated from imported cases, with DENV not being considered endemic [20]. Symptoms of DENV infection may include severe headache, severe joint and muscle pain, retro-orbital pain, nausea, and vomiting, and can result in dengue fever, dengue hemorrhagic fever, and dengue shock syndrome [21]. In 2022, a significant outbreak of JEV occurred on mainland Australia, marking an unprecedented event in a non-endemic region [22]. Before this occurrence, JEV had only been sporadically detected in Australia, primarily limited to the Torres Strait, Tiwi Islands (both situated off northern Australia), and occasionally in far northern Queensland. JEV, known for causing severe neurological symptoms and occasionally fatal outcomes, poses a significant risk to both humans and animals. MVEV, endemic to Papua New Guinea and Australia, is predominantly transmitted by *Culex* mosquitoes, with waterbirds as an amplifying host [15]. Similar to JEV, MVEV infection carries a low risk of clinical encephalitis (between 1:150 and 1:1000), but with case fatality rates ranging from 15% to 30% and long-term neurological complications in 30% to 50% of survivors [23]. WNV_KUN_, belonging to the same clade as JEV, is so far the only subtype of WNV found in Australia [24]. It was first identified in mosquitoes in northern Queensland in 1960 and has since been detected in all states of the country [25]. While WNV_KUN_ is primarily endemic to northern Australia, it spreads southward during periods of heavy rainfall, leading to increased waterbird populations and mosquitoes. Initially associated with only sporadic cases of non-fatal encephalitis in humans and horses, a notable shift occurred in 2011 when a substantial outbreak of neurological disease struck horses, with over 1000 equine cases in southeastern Australia [26]. Between 2000 and 2023, the annual average of reported WNV_KUN_ human cases was 1.83 within Australia [27]. In the last decade, ZIKV and CHIKV have also caused public health concern [28,29] since their principal mosquito vectors (*Aedes aegypti* or *Aedes albopictus*, the same vectors as for DENV) are currently established in tropical Queensland and the islands of the Torres Strait, respectively. While there have been no instances of local transmission in Australia, these viruses are circulating in Asia and Papua New Guinea, underscoring the potential risk they pose.

The global incidence of arboviral transmission through blood transfusion is relatively low when compared to mosquito-borne transmission; however, the risk persists due to the viremic phase preceding symptom onset, the prevalence of asymptomatic cases, and sometimes high infection rates observed during outbreaks [30]. The most prevalent arboviruses transmitted through transfusion are WNV and DENV [31]. Transfusion transmission is less common for RRV [32], JEV [33], and ZIKV [34,35], whereas transmission through transfusion for MVEV, BFV, and CHIKV has not been reported. The Australian Red Cross Lifeblood (Lifeblood) employs strict donor eligibility criteria and risk tolerability principles to strike a balance between minimizing risks to blood recipients and ensuring an adequate blood supply, and asks unwell donors to report post-donation to mitigate these risks [36].

Weather variability influences mosquito behavior and habitat suitability; hence, climate change in Australia and globally could result in variations in spatial and temporal distributions of these vectors, and the viruses they transmit [37]. Heavy wet-season rains and extensive flooding heighten the risk of arboviral disease [38]. Understanding the drivers of these diseases at a fine scale is also critical to understanding regional risks. For example, studies of RRV spatiotemporal variation predominantly focus on Queensland [39,40,41,42], with some also investigating regions of South Australia [43] or New South Wales [44]. However, this leaves many regions of the country largely overlooked. Likewise, studies of BFV spatiotemporal variation also predominantly focus on Queensland [45,46], despite cases of this disease being regularly notified in all States and Territories within Australia. The period of peak arboviral transmission varies for each virus and throughout the country. For example, in the Northern Territory, RRV is estimated as most active between December and March [47,48], and in the southern part of Western Australia, RRV is most active between September and May [49]. However, a comprehensive examination of spatiotemporal variation of different arboviruses across Australia as a collective entity remains incompletely understood. This knowledge gap necessitates a detailed investigation into the seasonal patterns of arboviruses throughout Australia. Additionally, certain arboviruses pose significant concerns due to their potential impact on the safety and supply of blood transfusion products, necessitating a greater in-depth understanding of spatiotemporal patterns.

This retrospective study aims to (i) assess the spatiotemporal distribution of eight arboviruses (BFV, CHIKV, DENV, JEV, MVEV, RRV, WNV_KUN_, and ZIKV) between 2002 and 2017 in Australia, (ii) identify high-risk clustering areas for the locally acquired arboviruses, (iii) identify climatic, spatial, and temporal factors that are associated with the incidence of each arboviral disease, and (iv) assess whether any arbovirus case numbers and weather variables affect the number of blood donations.

## 2. Materials and Methods

### 2.1. Study Area

This study included 311 Statistical Area level 3 (SA3s) boundaries, which comprise geographic Australia, according to the 2011 system of SA3 categorization [50] (Figure 1). The Australian Bureau of Statistics (ABS) has defined SA3 region categorization for regional data analysis at the regional level. SA3 regions are sufficiently small to enable detailed analysis of weather and climate data, yet broad enough to ensure inhabitant anonymity. The study included the states of New South Wales, the Northern Territory, South Australia, Tasmania, Queensland, Victoria, and Western Australia. Human research ethics approvals could not be secured for the Australian Capital Territory datasets due to logistic reasons, and thus this set was not included. As the Australian Capital Territory accounted for only a small proportion of arboviral notifications (0.2% of BFV or RRV cases, 1.3% of DENV cases, 0.8% of ZIKV cases, and 0% of MVEV, JEV, WNVKUN, or CHIKV cases), we do not consider that the exclusion is likely to affect the results of the study.

Reported cases were mapped to SA3 using the postcode of residence. Due to the absence of an exact one-to-one correspondence between postcodes and SA3 regions, the National Notifiable Diseases Surveillance System, Australian Government, Department of Health (NNDSS), used an ABS correspondence file from 2006 to generate a conversion file. This process enabled the determination of the SA3 region that predominantly corresponds to a single postcode, based on the highest proportion identified.

### 2.2. Data Source and Preparation

The case notification data for BFV, CHIKV, DENV, JEV, MVEV, RRV, WNVKUN, and ZIKV, spanning from January 2002 to September 2017, were sourced from the NNDSS. The dataset obtained represents every case within the study period that was diagnosed and subsequently notified to the NNDSS. Population estimates for SA3 areas for census years 2001, 2006, 2011, and 2016 were obtained from the ABS, with linear interpolation used for other years. Human population demographics by age group and sex were also collected from the ABS [51,52]. We then estimated the monthly incidence rates per 10,000 population.

Weather data spanning January 2002 to September 2017, including rainfall, mean monthly maximum and minimum temperatures, and mean monthly relative humidity, were obtained from the SILO database, maintained jointly by the Queensland Government and the Bureau of Meteorology [53]. Daily variation in temperature (DVT) was computed as the difference between mean monthly maximum and minimum temperatures. Lagged values for these weather variables were calculated at intervals of 1, 2, and 3 months.

Blood donation data from Lifeblood from 2007 to 2017 were obtained from the Lifeblood database. Earlier years were excluded due to incomplete data collection. The dataset includes monthly donation counts by donor demographic and location of residence at the SA3 level, along with annual counts of blood products obtained from individual donations, categorized by the type of donation. Monthly blood donation rates were calculated based on the ratio of blood donations to the number of potential donors aged 20 to 69 per 10,000 population within each SA3. A monthly time series dataset from January 2002 to September 2017 was compiled, assigning zero counts for periods with no reported cases. Two datasets were generated: (a) spanning 2002–2017, including SA3, states, year, month, virus name, transmission type, case count, incidence rate, human population, and weather variables with lag months, and (b) spanning 2007–2017, with the same variables but with the addition of blood donation rates per 10,000 population.

### 2.3. Case Definition

The current surveillance case definitions for disease notifications in Australia, accessible at: https://www.health.gov.au/casedefinitions (accessed on 23 May 2023), underwent significant revisions endorsed by the Communicable Diseases Network Australia in September 2003. Almost all jurisdictions adopted these new definitions by January 2004, with New South Wales following suit in August 2004 [54]. Before this national standardization, jurisdictions used a mix of 1994 National Health and Medical Research Council (NHMRC) case definitions, modified NHMRC-based definitions, and disease-specific definitions unique to their state. Notable changes to case definitions for BFV and RRV were implemented in January 2016, addressing concerns about overdiagnosis due to unreliable serological tests [54]. An ‘epidemic’ of false-positive IgM diagnoses of BFV emerged in October 2012, leading to the inclusion of both probable and confirmed cases in notifications since then. CHIKV infection became nationally notifiable in January 2015, with a separate case definition established in 2010 [54]. The designation of ZIKV, as a distinct, nationally notifiable disease from January 2016 onward, replaced the previous category of “flavivirus unspecified,” allowing for the notification of probable cases alongside confirmed cases [54]. The diagnosis month/year was obtained from the diagnosis date, a derived field reflecting the onset date or, in cases where the onset date is unknown, the earliest among specimen collection, notification, or notification received date/s. Additionally, the country of acquisition was supplied when accessible, indicating the suspected origin of the virus, whether acquired locally or overseas.

### 2.4. Data Analyses

#### 2.4.1. Exploratory Data Analysis and Visualization

To visualize temporal trends and average seasonal variations in the incidence rates of locally acquired and imported arboviruses, monthly mean incidence rates through the study period were calculated and plotted using RStudio software (2023.6.0.421) [55]. Spatiotemporal trends have been mapped using SA3s shapefile and QGIS (3.34) [56]. We examined the incidence rates per 10,000 population per state for the prevalent, locally acquired arboviruses, namely, BFV, RRV, and DENV.

#### 2.4.2. Seasonality of Arbovirus Cases

We plotted the incidence rate per 10,000 population of both locally transmitted and imported cases, by month, throughout the study period. We also performed a seasonal decomposition analysis to describe our time series dataset trends and seasonal factors for the locally acquired DENV, RRV, and BFV. The average incidence rate (IR) by 10,000 population was too small (below 0.01) for MVEV and WNV_KUN_ strains. Time series decomposition works by splitting a time series into three components: seasonality, trends, and random fluctuation. It helps reveal any underlying patterns and trends in the data, providing valuable insights for analysis and decision-making.

#### 2.4.3. Space–Time Cluster Analysis

This study employed the space–time scan statistic (SaTScan-v10.1.3) to achieve three objectives: (i) ascertain whether locally acquired arbovirus cases exhibited random spatial and temporal distribution, (ii) identify space–time clusters, and (iii) assess the statistical significance of the clusters. We used the Kulldorf method of retrospective space–time analysis with space–time permutation [57], and scanned for areas with high or low rates by year to determine the most probable cluster for each simulated dataset. The study employed SA3s in Australia (excluding the Australian Capital Territory) as spatial units (n = 311 SA3s in mainland Australia). Time was measured in calendar years. The cluster’s distribution and significance were assessed through Monte Carlo replication sets under the null Hypothesis (H0), with replications greater than 999 for adequate power analysis [57].

**H0.** *No statistically significant space–time clusters exist*.

The *p*-value cutoff was 0.05. Observed, expected cases, observed/expected cases ratios, *p*-values, and Monte Carlo rank were reported. The observed/expected cases ratio is the estimated risk within the cluster divided by the estimated risk for the study region [58]. SaTScan software (version 10.1.3) used a default maximum spatial cluster size of 50% of the population at risk to detect large clusters with potentially low relative risk but high statistical significance, while the maximum temporal cluster size was set at 50% of the study period. Visualization of statistically significant clusters was conducted using QGIS (version 3.34.5).

#### 2.4.4. Collinearity Check and Correlation Analysis

Multicollinearity occurs when the independent variables in a regression model exhibit high levels of correlation with each other. This phenomenon complicates the interpretation of the model and can lead to overfitting issues. Correlation, a statistical metric, gauges the degree to which two or more variables fluctuate in tandem. The correlation coefficient, r, ranges between +1 and −1, signifying the strength of the relationship. A positive correlation indicates that the variables increase or decrease concurrently, while a negative correlation means that one variable rises as the other falls. We estimated (i) the variance inflation factor (VIF) from the vif() function in the car package in R, which measures the influence of collinearity on the variance of our coefficient estimates, and (ii) r, which tells the strength and direction of the association between the two variables.

#### 2.4.5. Model Building and Evaluation

To assess the association between climate, spatial, temporal, and population demography factors with the following outcome variables separately, we focused on four different approaches:(1)count~Transmission_Name, where ‘Transmission_Name’ is the variable defining the locally acquired arbovirus (dataset in long format),(2)count for each arbovirus (dataset in wide format),(3)counts of arboviruses in SA3s identified in the cluster analysis,(4)blood donation number,(5)blood donation number in SA3s identified in the cluster analysis.

Analyzing the data in a long format allowed us to investigate the association between ‘Transmission_Name’ and overall count, while analyzing it in a wide format allowed for a more detailed examination of individual virus counts. We developed Poisson and negative binomial regression models to investigate the associations in (1), (2), and (4). We employed the Poisson and negative binomial regression and generalized estimating equations (GEE) approach to model the incidence of arbovirus cases (3) and blood donation numbers (5). Poisson and negative binomial regressions are typically used to model count data. However, sometimes, it is more relevant to model rates instead of counts. Hence, we added the log of population as an offset, which means that we looked at expected case numbers per member of the population. GEE is a statistical method suitable for analyzing correlated data, such as repeated measures or clustered data, which may arise when studying disease incidence or counts in different geographic areas over time [59]. To assess the relative goodness of fit of different models while penalizing for model complexity, we compared the values across models and selected the model with the lowest Akaike Information Criterion (AIC) and Bayesian Information Criterion (BIC).

### 2.5. Ethical Considerations

This study was approved by the Lifeblood Ethics Committee, Australian Red Cross Lifeblood (approval number: Viennet 12012021b).

## 3. Results

### 3.1. Distribution of Arboviruses in Space and Time

During 2002–2017 in mainland Australia, the arbovirus with the highest number of locally acquired cases was RRV, followed by BFV and DENV, while the arbovirus with the highest number of reported imported cases was DENV, followed by CHIKV (Table 1).

We present here the spatiotemporal distribution of the three most prevalent arboviruses in humans in Australia (RRV, BFV, and DENV). The spatiotemporal distribution of cases of MVEV (locally acquired), WNV (WNV_KUN_ locally acquired and WNV imported), JEV (imported), ZIKV (imported), DENV (imported), and CHIKV (imported) is provided in Appendix A.

RRV incidence rates were also scattered through space and time (Figure 2a), with most cases acquired in Queensland (45%), Western Australia (16%), and New South Wales (15%; Figure 2b). Peak transmission of RRV occurred almost every year, with the highest also following a two- to three-year cycle (in 2003, 2006, 2008, 2010, 2015, and 2017; Figure 2c).

BFV incidence rates were relatively scattered through time and space (Figure 3a), with approximately 55% of BFV cases occurring in Queensland, 19% in New South Wales, and 10% in NT (Figure 3b). Peaks of transmission occurred every two to three years (2003, 2006, 2008, and 2013; Figure 3c).

Locally acquired DENV incidence rates were limited to north and central Queensland (Figure 4a,b), and the years of peak transmission were 2003, 2009, and 2013 and declined over time from 2002 to 2017 (Figure 4c).

The two SA3s with the highest incidence rate per 10,000 population were Port Douglas–Daintree and Innisfail–Cassowary Coast (both in Queensland) for BFV, Litchfield (the Northern Territory) and Kimberley (Western Australia) for RRV, and Cairns–South and Innisfail–Cassowary Coast (both in Queensland) for DENV. These findings are in agreement with the results of the cluster analysis in Section 3.4.

### 3.2. Seasonality of Locally Acquired and Imported Cases

The incidence of both locally transmitted and imported cases displayed discernible seasonal patterns. Overall, locally acquired cases of all arboviruses were less common during winter, with increasing occurrences during summer and autumn (Figure 5a), whereas imported cases were observed throughout the year (Figure 5b). RRV exhibited an extended seasonal transmission pattern, spanning from October to June in Western Australia and New South Wales. Conversely, in the Northern Territory, the transmission pattern of both BFV and RRV was unclear. In Queensland, RRV transmission seemed to peak from January to May; however, transmission appeared to be sustained over winter, as for BFV. The season of local transmission of DENV typically occurred during the austral warmer months, spanning from late spring through summer and into early autumn.

Seasonal decomposition analysis showed that the locally acquired arboviruses (DENV, RRV, and BFV) displayed seasonality in each state (Appendix A). The magnitude of the seasonal component relative to the trend and residual components determines if there is strong seasonality. If the seasonal component is large compared to the other components, then there is likely strong seasonality in the data. Conversely, if the seasonal component is small compared to the other components, then the seasonality in the data is likely weak. The analysis revealed a decreasing trend for DENV infections in Queensland (Appendix A) and BFV infections in New South Wales (Appendix A) and a rising trend in RRV infections in Queensland, New South Wales, Western Australia, and Victoria, with consistent and smooth seasonality curves observed in locally transmitted cases across the study period (Appendix A, Appendix A, Appendix A, and Appendix A, respectively).

### 3.3. Origin of Imported Cases

The imported cases of DENV, CHIKV, and JEV were mainly from infections within the continent of Asia, followed by Oceania (Figure 6), with Indonesia being the first source of origin of the above arboviruses (representing 53%, 34%, and 38% of the total count for DENV, CHIKV, and JEV, respectively; Table 2). The total number of DENV imported cases consistently peaked above 100 imported cases from the end of 2010. Additionally, CHIKV began to be regularly imported from 2010 onwards.

### 3.4. Spatiotemporal Clustering of Locally Acquired Arboviruses

Three clusters were identified for BFV, three for RRV, three for DENV, two for WNV_KUN_, and four for MVEV during the study period (Appendix A). All clusters were significant (H0 was rejected), except for MVEV (*p*-values > 0.05).

For BFV (Figure 7a), the first cluster encompassed part of the Northern Territory and Western Australia (from 01/2013 to 12/2013, n = 1355, obs./exp. = 2.18, *p*-value < 1 × 10^−17^). The second BFV cluster contained South Australia, New South Wales, Tasmania, and Victoria (from 01/2013 to 12/2013, n = 567, obs./exp. = 0.43, *p*-value < 1 × 10^−17^), while the third BFV cluster contained part of north and central Queensland (from 01/2016 to 12/2016, n = 114, obs./exp. = 2.2, *p*-value < 76 × 10^−17^). For DENV (Figure 7b), the first cluster encompassed part of Northeast Queensland (from 01/2011 to 12/2011, n = 51, obs./exp. = 10.39, *p*-value < 1 × 10^−17^). The second DENV cluster also contained part of Northeast Queensland (from 01/2013 to 12/2013, n = 30, obs./exp. = 30.11, *p*-value < 47 × 10^−17^), while the third DENV cluster contained part of Northeast Queensland (from 01/2009 to 12/2009, n = 121, obs./exp. = 21.12, *p*-value < 16 × 10^−9^). For RRV (Figure 7c), the first cluster encompassed part of South Australia (from 01/2011 to 12/2011, n = 2037, obs./exp. = 4.46, *p*-value < 1 × 10^−17^). The second RRV cluster contained a small part of Southeast Queensland (from 01/2015 to 12/2015, n = 4202, obs./exp. = 2.05, *p*-value < 1 × 10^−17^), while the third RRV cluster contained part of the Northern Territory and Western Australia (from 01/2012 to 12/2014, n = 4970, obs./exp. = 1.66, *p*-value < 1 × 10^−17^). For WNV_KUN_ (Figure 7d), the first cluster encompassed most of Queensland and part of the Northern Territory (from 01/2010 to 12/2017, n = 0, obs./exp. = 3.51, *p*-value = 0.013). The second WNV_KUN_ cluster contained South Australia, Western Australia, part of the Northern Territory, and Queensland (from 01/2003 to 12/2010, n = 0, obs./exp. = 3.11, *p*-value = 0.048). Finally, for MVEV, the first cluster contained part of the Northern Territory and Queensland (from 01/2015 to 12/2015, n = 2, obs./exp. = 2.86, *p*-value = 0.53). The second MVEV cluster contained part of South Australia, Western Australia, and the Northern Territory (from 01/2002 to 12/2005, n = 0, obs./exp. = 2.61, *p*-value = 0.73). The third MVEV cluster contained part of South Australia, the Northern Territory, Queensland, and New South Wales (from 01/2005 to 12/2008, n = 2, obs./exp. = 1.96, *p*-value = 0.96). Finally, the last MVEV cluster contained part of Western Australia and the Northern Territory (from 01/2011 to 12/2015, n = 0, obs./exp. = 1.86, *p*-value = 0.98). Because of the generally large size of these clusters (except for DENV clusters and RRV cluster 2), identifying the spatial specificity of an outbreak was limited.

Overall, the most likely clusters were DENV clusters 1 (Innisfail–Cassowary Coast (Queensland) in 2011, containing one SA3, nSA3 = 1) and 2 (Port Douglas–Daintree (Queensland), Cairns (Queensland) in 2013, nSA3 = 2), MVEV clusters 1 (Katherine, Barkly (Western Australia) in 2015, nSA3 = 2) and 3 (Darwin region (the Northern Territory) in 2011–2015, nSA3 = 6), RRV cluster 1 (South Australia in 2011, nSA3 = 48), and BFV cluster 3 (north Queensland in 2016, nSA3 = 12). Most of the clusters occurred from 2011 to 2015.

### 3.5. Statistical Analyses

#### 3.5.1. Arbovirus Transmission and Associations

We checked for correlation and multicollinearity and found that average maximum temperature, average relative humidity at maximum temperature, and their lag months were problematic variables (Appendix A); hence, we removed these variables from the analyses on the count of arboviruses and defined this resulting model as the ‘base model’ (Appendix A).

In our study, we initially employed a general modeling approach to investigate the relationship between locally acquired arbovirus cases and population number, various weather, and spatiotemporal variables. We treated ‘Count’ as the outcome variable and considered ‘Transmission_Name’ as an independent variable, alongside other predictors. Subsequently, we developed models for each specific arbovirus.

The optimal model from the initial general approach was a full negative binomial model, incorporating ‘Transmission_Name’, the ‘base model’, population (offset), month–year, and SA3s (Appendix A). Notably, the early months of the year (January to April) were consistently significantly associated with a higher count of arbovirus cases. When comparing different regions, Cairns–South (Queensland) exhibited a significantly higher expected log count of cases, whereas Botany (New South Wales) showed a significantly lower expected log count of cases, compared to the reference SA3 (Adelaide–City) of all arboviruses combined. Various meteorological factors, such as average rainfall and its lag at 1 and 2 months, average minimum temperature, and its lag at 2 and 3 months, daily variation in temperature, and relative humidity at minimum temperature, and its lag at 2 and 3 months, were all found to be significantly and positively associated with the count of arbovirus cases. However, the magnitude of change in the outcome variable with a one-unit increase in the independent variable was negligible (around a factor of 1).

Subsequent models developed for each locally acquired arbovirus specifically revealed different optimal models based on AIC and BIC criteria:(i)For locally acquired MVEV and WNV_KUN_ cases, the optimal model was a Poisson regression, with only the meteorological variables (defined as the ‘base model’).(ii)For locally acquired DENV cases, the optimal model was a negative binomial with a base model.(iii)For locally acquired BFV and RRV cases, the optimal model was the full model (base model, month–year, offset of log (population), and SA3s).

We found that rainfall played a significant role in the transmission of BFV, DENV, and RRV. Additionally, the average minimum temperature appeared as a crucial variable influencing the transmission of BFV, RRV, MVEV, and locally acquired cases of DENV. Relative humidity was identified as a key factor for RRV, MVEV, and locally acquired cases of WNV_KUN_, with its lag months serving as important predictors for locally acquired DENV, RRV, and BFV. Moreover, daily temperature variation was a significant predictor for BFV, RRV, and MVEV. BFV and RRV count models exhibited similar trends, with significant associations observed with month/year, average rainfall and its lag at 1 month, average minimum temperature, DVT, and lagged relative humidity at minimum temperature (2 and 3 months). However, the direction and magnitude of these associations differed between the two arboviruses, except for average rainfall, which showed a significantly negative association with BFV and RRV count. For each one-unit increase in rainfall (mm), the expected log count of BFV and RRV counts decreased by 0.0003 and 0.0002, respectively, which is extremely negligible. Detailed results for each arbovirus can be found in Appendix A.

Finally, we conducted separate analyses for the counts of locally acquired BFV, RRV, and DENV cases in distinct regions highlighted as important clusters: Port Douglas–Daintree, Innisfail–Cassowary Coast (BFV), Litchfield and Kimberley (RRV), Cairns–South, and Innisfail–Cassowary Coast (DENV), using Poisson, negative binomial, and GEE approaches. The best models were from using GEE approaches. Our analysis revealed compelling associations between environmental factors and the number of arbovirus cases in these regions. In Port Douglas–Daintree, we observed a significant and negative correlation between the number of BFV cases and the daily variation in temperature. In Innisfail–Cassowary Coast, the daily variation in temperature had a significant negative association, and average rainfall exhibited a significant positive association with BFV number of cases (*p* < 0.05). Similarly, in Litchfield, the number of RRV cases showed a significant and negative association with the daily variation in temperature, while in the Kimberley, the average relative humidity at minimum temperature and average rainfall were significantly associated, positively and negatively, respectively, with RRV number of cases (*p* < 0.05). For Innisfail–Cassowary Coast, our analysis indicated a significant and positive association between the number of cases of DENV and both the average relative humidity at minimum temperature and average rainfall. In contrast, in Cairns–South, significant associations were found between the number of cases of DENV and both the average minimum temperature and daily variation in temperature (*p* < 0.05). For comprehensive insights, readers are referred to Appendix A.

#### 3.5.2. Blood Donation Numbers and Associations

Based on AIC and BIC values, the best model with blood donation numbers as the outcome variable was the negative binomial model, as follows:

donorsnumber~Virus_name* IR10,000 + MonthYear +Rainavg + meanMaxTavg + meanMinTavg +DVT + meanRHTMaxavg + meanRHTMinavg + lag1MRainavg + lag1MmeanMaxTavg + lag1MmeanMinTavg + lag1MmeanRHTMaxavg + lag1MmeanRHTMinavg + lag2MRainavg + lag2MmeanMaxTavg + lag2MmeanMinTavg + lag2MmeanRHTMaxavg + lag2MmeanRHTMinavg + SA3_NAME11 + offset(log(popagedonor)), where ‘IR10,000’ is the incidence rate per 10,000 population, and ‘popagedonor’ is the number of people in a given SA3 in the age of donation [19,20,21,22,23,24,25,26,27,28,29,30,31,32,33,34,35,36,37,38,39,40,41,42,43,44,45,46,47,48,49,50,51,52,53,54,55,56,57,58,59,60,61,62,63,64,65,66,67,68,69].

The best model included weather variables and their lags. Surprisingly, the incidence rate of arbovirus cases per 10,000 population alone was significantly and positively associated with the number of blood donations collected (*p* < 0.05) in all SA3 considered, although it was extremely small. However, the interaction between the incidence rate and the local transmission of DENV was negatively and significantly associated with the number of blood donations (*p* < 0.05). An increase of one unit in the incidence rate when DENV was considered corresponded to a factor of exp(−0.025) = 0.97 change in the number of blood donations. Weather variables, such as average rainfall, the average relative humidity at maximum and minimum temperature, as well as their lag at 1 and 2 months, were also significantly associated with the number of blood donations (*p* < 0.05). All results are presented in detail in Appendix A.

Finally, we conducted separate analyses for the number of blood donors in the distinct regions identified as important clusters in Section 3.4 using Poisson, negative binomial, and GEE regressions. At the SA3 level, focusing on Port Douglas–Daintree and Innisfail–Cassowary Coast (clusters detected for BFV), Litchfield and Kimberley (clusters detected for RRV), and Cairns–South and Innisfail–Cassowary Coast (clusters detected for DENV), the incidence rate of the given arbovirus was not significantly associated with the number of blood donations in this given SA3. However, we found that the average relative humidity at minimum temperature was significantly and positively associated with the number of blood donations in Litchfield (*p* < 0.05). In Kimberley, the average relative humidity at minimum temperature and the daily variation in temperature were significantly associated (negatively and positively, respectively) with the number of blood donations (*p* < 0.05). Finally, the average relative humidity at minimum temperature was significantly and negatively associated with the number of blood donations in Cairns–South (*p* < 0.05). No significant associations were observed in Innisfail or Port Douglas–Daintree. The significant associations were particularly small. For comprehensive insights, readers are referred to Appendix A.

## 4. Discussion

The identification of environmental patterns and parameters influencing virus transmission, alongside the spatiotemporal dynamics of disease spread, is essential for understanding, predicting, and mitigating future arbovirus transmission globally, as well as in Australia. Several articles have explored spatiotemporal trends in arbovirus transmission, but their focus has often been limited to a single virus or state. For instance, Murphy et al. [39] conducted a study in Queensland, revealing that the highest incidence rates were concentrated in predominantly rural suburbs north of Brisbane City, with notable hot spots in peri-urban areas where residential, agricultural, and natural land use types intersected. Another study, albeit older, identified an association between climate variability and the transmission of RRV infection in Townsville, a coastal region in northern Queensland, explaining 30% to 37% of the variance, with a significant albeit low power of association [60]. Additionally, previous research has highlighted the sensitivity of RRV transmission cycles to climate and tidal variability [61]. Factors such as rainfall, temperature, and high tides were identified in other work as major determinants of RRV transmission at a macro-level, although the nature and magnitude of their interrelationship with climate variability, mosquito density, and transmission varied across geographic areas and socioenvironmental conditions [61]. Both RRV [41] and BFV [45] transmission exhibit seasonal patterns, with peak transmission expected from February to May, although slight variations exist across the country. Recent shifts in climate and globalization have also raised the alarm about JEV’s potential for emergence and spread in Australia and worldwide [62]. A modeling study suggested that climatic conditions in January and March 2022 in Victoria, Australia, notably the minimum temperature and precipitation needed for efficient JEV transmission, likely led to higher-risk conditions that were more favorable for JEV transmission in the lead-up to that year’s outbreak of JEV in Victoria [63]. A range of climatic conditions favorable for MVEV transmission was also observed before the MVEV outbreak in Victoria in 2023, including higher water levels in a major river system and extensive flooding [64].

The present study elucidated the spatial patterns and temporal trends of arbovirus transmission nationally within Australia over 16 years. The spatial distribution analysis revealed varying patterns of arbovirus transmission across different states. Queensland is an important hotspot for locally acquired MVEV, RRV, BFV, and DENV cases, reinforcing the need for targeted interventions and control strategies in this region. The lack of defined seasonal transmission patterns in the Northern Territories underscores the complexity of arbovirus ecology and the potential for overwintering of vectors—hence a lengthy transmission period—necessitating further research into local environmental drivers. Our findings appear to mirror patterns noted in earlier research [65,66,67]; however, with a finer spatial resolution and over an extended study period. The temporal analysis unveiled a declining trend in reported cases of DENV infections in Queensland and BFV cases in New South Wales, contrasting with a rising trend in reported cases of RRV across Queensland, New South Wales, Western Australia, and Victoria. The declining number of DENV-reported cases is most likely linked to the World Mosquito Program’s long-lasting *Wolbachia* method undertaken in 2011 in Cairns and surrounding locations [68,69]. The outbreak of RRV in 2015 could have resulted from a confluence of ecological factors, potentially driven by above-average rainfall, leading to increased abundance of mosquito populations, as suggested by Jansen et al. [70].

Notably, RRV and BFV exhibited distinct seasonal peaks, with RRV transmission occurring almost annually and BFV exhibiting peaks every two to three years. These patterns are crucial for public health planning, resource allocation, and implementing targeted control measures. The identification of extended seasonal transmission of RRV from September to July in Western Australia and New South Wales highlights the importance of year-round vigilance and surveillance. *Culex annulirostris*, one of the main vector species of RRV and BFV, inhabits various habitats across the country, including coastal and inland regions, where it thrives in diverse environments, such as vegetated freshwater pools, both temporary and permanent, including natural and human-made wetlands [71]. It demonstrates rapid reproductive rates in warmer temperatures and can enter a dormant state during colder temperatures, prolonging its lifespan. This dormancy period [72], coupled with its opportunistic feeding behavior, is believed to play a significant role in the transmission cycle of RRV and BFV in Australia [73]. Therefore, from a preventive point of view, it is crucial to increase public awareness of the risk between September and July, if not all-year-round.

Throughout the study period, the season of local transmission of DENV typically occurred during the warmer months, spanning from late spring through summer and into early autumn. However, the number of reported DENV cases declined over time from 2002 to 2017 due to the introduction of the *Wolbachia* program targeting the *Ae. aegypti* population [74], unlike the number of DENV-imported cases, which reached regular peaks above 100 imported cases.

Seasonal decomposition analysis unveiled distinct seasonal patterns for the primary locally acquired arboviruses (DENV, RRV, and BFV) across various states, with varying trends observed: DENV infections decreased in Queensland, BFV infections decreased in New South Wales, and RRV infections increased in Queensland, New South Wales, Western Australia, and Victoria. Overall, locally transmitted cases of RRV exhibited a consistent and smooth seasonality curve throughout the study period in these regions. We did not assess the seasonal decomposition of MVEV and WNV_KUN_ because their average incidence rate per 10,000 population was too low to conduct a meaningful time series decomposition analysis.

The main source of importation of arbovirus cases was the continent of Asia, with Indonesia predominantly reported as a country of source, which has already been described for DENV [20,74,75,76,77] and CHIKV [76,78]. Increased movement and proximity to endemic regions present an important risk for countries with endemic vectors, such as Australia [79,80]. The sharp decrease in international passenger flights due to COVID-19 mitigation measures in 2020 resulted in a decline of over 70% in imported dengue cases in Queensland compared to the preceding five years. This substantial reduction in flights mitigated a risk pathway for importation through viremic travelers and stowaway mosquitoes [81]. It is important to note that the risk posed by importation via sea cargo remained unaffected [81]. For effective arbovirus-control programs targeting viruses such as DENV, JEV, WNV, CHIKV, and ZIKV in Australia, it is imperative to integrate insights from the epidemiology of these viruses in other countries. This includes strategies to minimize virus importation, enhance control measures for cryptic larval habitats, and actively involve the community in efforts to reduce vector breeding, as highlighted by Viennet et al. [82].

The identification of spatial and temporal clusters of arbovirus cases enhances our understanding of disease dynamics and can guide targeted interventions and resource allocation. We effectively pinpointed clusters of BFV, DENV, RRV, and MVEV transmission, with most of the clusters occurring from 2011 to 2015. One explanation could be the occurrence of a La Niña event, which started in late 2010 and continued into early 2011 [83]. El Niño events occurred in late 2002 to early 2003, late 2006 to early 2007, and between 2015 and 2016 [84]. El Niño*-*Southern Oscillation (ENSO), with two extremes, El Niño and La Niña*,* is related to various health outcomes, including waterborne disease, vector-borne disease, and natural-disaster-related deaths [85]. While El Niño events typically bring drier and warmer conditions to eastern Australia, which might reduce mosquito breeding habitats and potentially decrease arbovirus transmission, La Niña events, characterized by wetter and cooler conditions, may create more favorable breeding environments for mosquitoes and increase the risk of arbovirus transmission. For example, Qian et al. [86] found a negative association between El Niño events and RRV incidence in Redlands (Queensland). During La Niña events, increased rainfall and flooding can lead to the proliferation of mosquito populations, potentially increasing the transmission of arboviruses, such as DENV [87], RRV [88], and BFV, in affected regions. However, the relationship between ENSO events and arbovirus transmission in Australia can be complex and is influenced by various factors, including local environmental conditions, mosquito ecology, and human behavior. Therefore, while ENSO events can be a contributing factor to arbovirus transmission dynamics, they may not always directly correlate with outbreaks, and other factors must also be considered.

Applying statistical models to investigate the relationship between arbovirus cases and meteorological, demographic, and spatiotemporal variables provides valuable predictive insights. The past three decades have witnessed extensive documentation of environmental and climatic factors influencing the transmission of RRV across Australia. Studies have consistently underscored the significance of various environmental determinants, such as precipitation, temperature, tides, floods, and humidity, in the incidence of RRV notifications [40,61,89,90]. Notably, in our study, rainfall was an important variable for BFV, DENV, and RRV transmission, as it has already been shown through several studies for RRV [90,91,92,93,94,95], BFV [96,97], and DENV [65,98,99,100,101]. The average minimum temperature is an important variable for BFV, RRV, MVEV, and locally acquired DENV. The relative humidity is an important variable for RRV, MVEV, and WNV_KUN_ (locally acquired), while its lag months are important predictors for locally acquired DENV, RRV, and BFV. Relative humidity at a lag time may facilitate the mosquito lifecycle and vegetation growth, thus indirectly contributing to disease transmission [44,102]. Finally, daily variation in temperature was an important predictor for BFV, RRV, and MVEV. Daily variation in temperature is rarely included in analyses, while we showed that this is an important predictor for at least four arboviruses transmitted in Australia. Predictors’ importance depended on the spatial scale of the analysis. Specifically, once we focused on the predicted clusters, indicative of a finer geographical scale, we observed that average rainfall, average minimum temperature, daily variation in temperature, and average relative humidity were the main predictors. Our study, utilizing GEE approaches, identified significant associations between environmental factors and arbovirus incidence at a fine scale, including negative associations between BFV cases and temperature variation in Port Douglas–Daintree, and positive correlations between DENV cases and rainfall in Innisfail–Cassowary Coast.

This study is the first to investigate the relationship between blood donations and arboviruses of significance to Australia over 16 years, highlighting the potential, albeit extremely small, impact of DENV transmission on blood donation numbers. We found that the number of blood donations was significantly associated with the incidence rate of all viruses per 10,000 population and its interaction with local transmission of DENV considering all SA3s, as well as some weather variables, such as the average rainfall and the average relative humidity at maximum and minimum temperature. However, at the specific SA3 level, focusing on detected clusters, no significant association was observed between the incidence rate of a locally acquired arbovirus and the number of blood donations. The average relative humidity at minimum temperature emerged as the primary variable associated with the number of blood donations in Litchfield, Kimberley, and Cairns–South. Higher humidity levels during cooler temperatures might influence donor behavior, potentially increasing (e.g., in Litchfield) or decreasing (e.g., in Kimberley and Cairns–South) the number of donations during certain times of the year. However, providing a plausible explanation for these findings is challenging, as the relationship between environmental factors and donation patterns is complex and may be influenced by a variety of socioeconomic, behavioral, and physiological factors that were not accounted for in this analysis.

Lifeblood has implemented donor deferral policies, including a six-month deferral from the date of recovery for donors reporting encephalitis, a four-week deferral for recipients of live JEV vaccines, and a four-month deferral for donors recovering from flavivirus infections, while those who have visited areas with known outbreaks are deferred for four weeks after leaving the risk-exposure area. During local DENV outbreaks, donations from affected areas are limited to plasma product production.

The present study has several limitations. Firstly, the notification dataset was extracted from the NNDSS on a specific date, and due to the dynamic nature of the NNDSS, retrospective revisions may occur, leading to potential variations compared to published reports. Moreover, notification data represent only a portion of the total cases in the community, reflecting instances where healthcare was sought, a test conducted, and a diagnosis made, followed by notification to health authorities. Not all cases manifest symptoms, and not all symptomatic cases seek medical care. Consequently, the extent of underrepresentation across all cases remains unknown and likely varies by disease and jurisdiction. Interpreting these data requires caution, as changes in notifications over time may not solely reflect disease prevalence or incidence changes. Factors such as shifts in testing policies, screening programs targeting high-risk populations, advancements in diagnostic tests, and public awareness campaigns can influence annual notification rates. Additionally, limitations include potential underreporting and misreporting of autochthonous and overseas-acquired cases due to asymptomatic infections. Furthermore, discrepancies may arise between the residential address provided during notification and the actual location of infection, especially during holidays, posing challenges in accurately pinpointing disease transmission locations. Long-format statistical analysis might overlook the effects of a specific arbovirus, while wide-format statistical analysis could lack efficiency in modeling the relationship between ‘Transmission_Name’ and count. In this context, the lack of efficiency in modeling would result in less accurate predictions, increased computational time, and potentially misleading conclusions due to the inability to capture the true dynamics of the data. The spatial resolution of the study, which used fine geographical levels but not the finest, may miss some local variations (microclimate) due to broader spatial aggregation. This could lead to an underestimation of localized outbreaks or variations in arbovirus transmission patterns. The temporal scope of the analysis, spanning from 2002 to 2017, may not fully capture recent trends or shifts in risk factors that could influence current and future transmission dynamics. Changes in climate, vector, and human behavior, and in public health interventions over time, may have altered the patterns of arbovirus spread. Additionally, while the study focused on eight significant arboviruses, which is quite extensive, there may still be other arboviruses not included in the analysis that could impact public health and contribute to the overall burden of arboviral diseases in Australia. Finally, the datasets used herein were gathered before the COVID-19 pandemic. While not a focus of the present study, the pandemic might have impacted extrapolation from the trends in 2002–2017 to more recent years. Indeed, the COVID-19 pandemic had diverse effects on arbovirus transmission in Queensland, Australia. A three-fold increase in RRV cases in April–May 2020 was observed, with cases suspected to have been exposed closer to their residences. In contrast, a decrease in imported cases of DENV was observed during the first half of 2020, due to reduced international travel to Australia [81].

Our findings underscore the confinement and decline of DENV risk at the local level, while highlighting the widespread distribution of BFV, MVEV, and RRV risks, covering expansive regions due to the widespread distribution of their principal mosquito vectors and ability to overwinter in certain regions. Integrating cluster detection into disease surveillance enables the identification of high-risk areas for targeted interventions. The significance of these clusters, particularly for BFV, RRV, and DENV, highlights potential areas of heightened transmission and the need for localized control measures. The proposed establishment of a communicable disease center in Australia is pivotal for effectively managing and mitigating future disease risks and breaking down silos among states. With progress toward this initiative now closer than ever before [103], initiatives such as Lifeblood’s heightened surveillance and emphasis on reporting post-donation illnesses, especially in outbreak-affected regions, underscore the importance of proactive measures in disease control.

## 5. Conclusions

To control or prevent future outbreaks, mosquito and disease surveillance, along with public health measures, should continue to target areas where mosquito vectors are established. Particular attention should be paid to BFV, MVEV, and RRV transmissions occurring in nearly all states and territories. Surveillance efforts should be intensified during the wet season, but also from October to June in Western Australia, and year-round in Queensland for RRV. This study contributes to understanding the intricate relationships among disease cases, weather variables, and spatiotemporal factors at a fine geographical scale, marking the first comprehensive examination of eight important arboviruses across Australia over a 16-year period. The fine-scale analysis allowed for the identification of specific geographic areas susceptible to arbovirus transmission, enabling more precise public health responses tailored to local conditions, ultimately enhancing disease surveillance and control efforts. Effective management of these public health threats requires multidisciplinary collaboration among meteorologists, entomologists, healthcare professionals, epidemiologists, and biostatisticians to improve understanding and response strategies.

## Figures and Tables

**Figure 1 biology-13-00524-f001:**
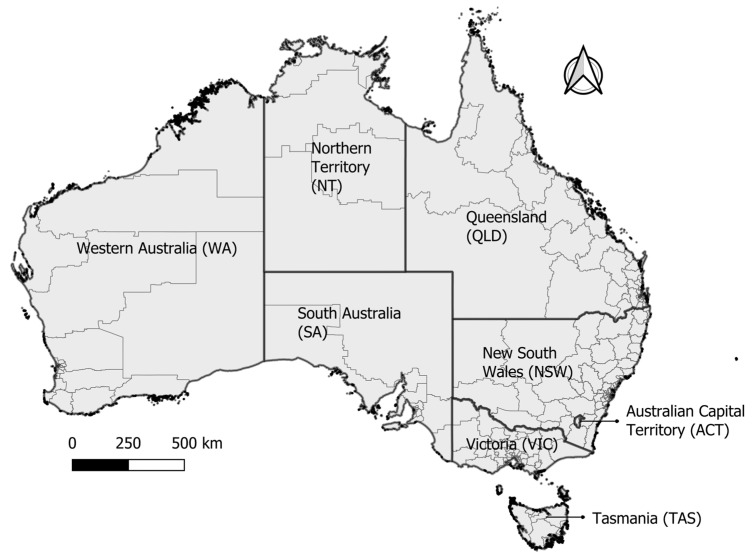
Map of Australia with states. Statistical Area level 3 units are shown in grey borders.

**Figure 2 biology-13-00524-f002:**
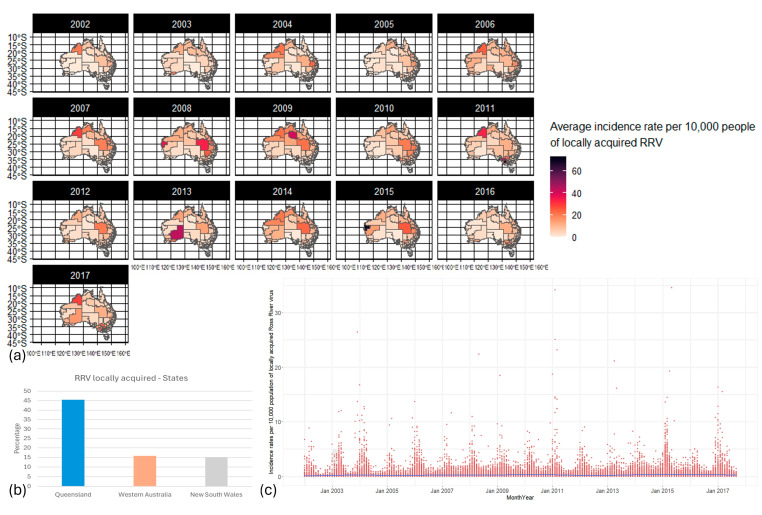
Locally acquired Ross River virus distribution from 2002 to 2017 in Australia. (**a**) Maps of the distribution of RRV incidence rates per 10,000 population per year through SA3s. (**b**) The local incidence rate (IR), as a percentage, of RRV proportion, by state. (**c**) Distribution of RRV incidence rate per 10,000 population through Australia. Red dots: incidence rate per 10,000 population per time and space; blue line: smoothed line to the plot. By default, geom_smooth() in RStudio (2023.6.0.421) [55] uses a Loess smoothing method for small datasets and a generalized additive model for larger datasets.

**Figure 3 biology-13-00524-f003:**
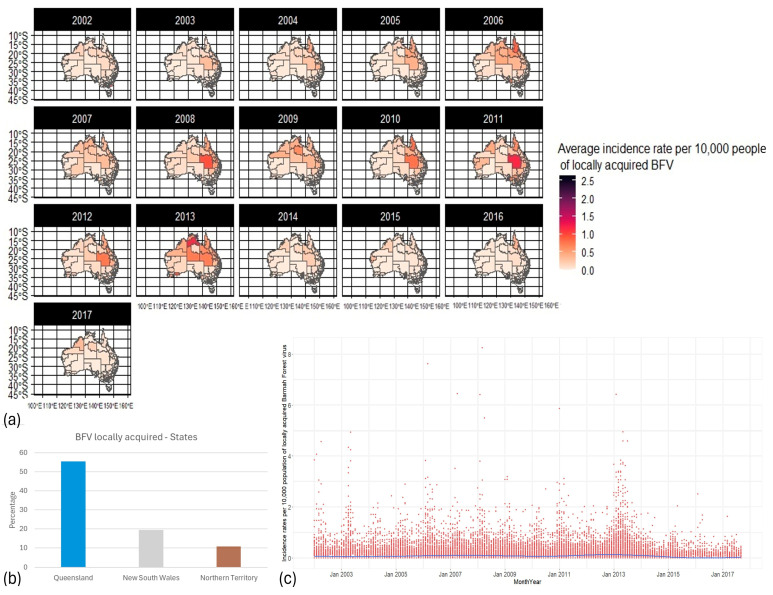
Barmah Forest virus distribution from 2002 to 2017 in Australia. (**a**) Maps of the distribution of BFV incidence rates per 10,000 population per year through SA3s. (**b**) The local incidence rate (IR), as a percentage, of BFV proportion, by state. (**c**) Distribution of BFV incidence rate per 10,000 population through Australia. Red dots: incidence rate per 10,000 population per time and space; blue line: smoothed line to the plot. By default, geom_smooth() in RStudio (2023.6.0.421) [55] uses a Loess smoothing method for small datasets and a generalized additive model for larger datasets.

**Figure 4 biology-13-00524-f004:**
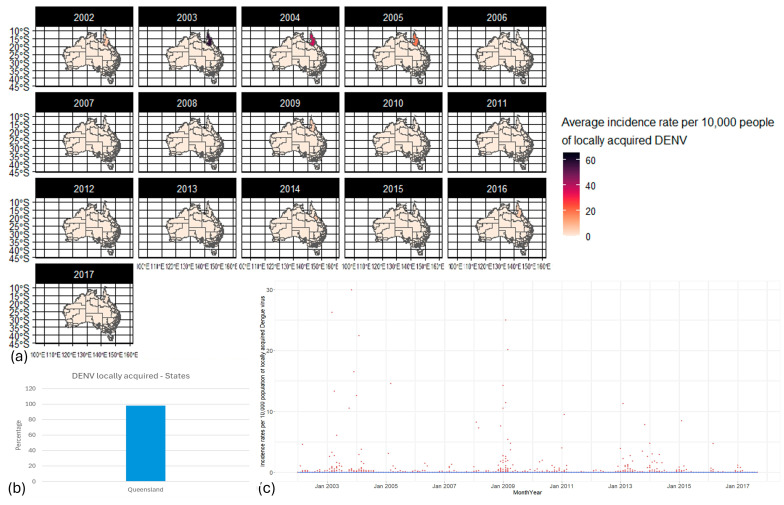
DENV distribution from 2002 to 2017 in Australia. (**a**) Maps of the distribution of DENV incidence rates per 10,000 population per year through SA3s. (**b**) The local incidence rate (IR), as a percentage, of DENV proportion, by state. (**c**) Distribution of DENV incidence rate per 10,000 population throughout Australia. Red dots: incidence rate per 10,000 population per time and space; blue line: smoothed line to the plot. By default, geom_smooth() in RStudio (2023.6.0.421) [55] uses a Loess smoothing method for small datasets and a generalized additive model for larger datasets.

**Figure 5 biology-13-00524-f005:**
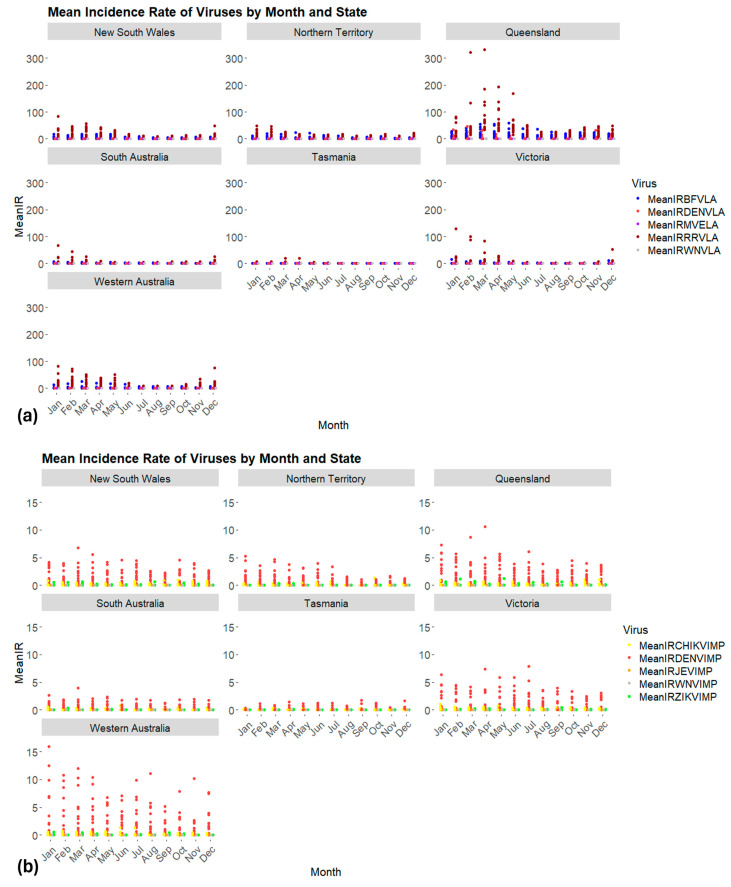
Monthly incidence rates per 10,000 population of eight arboviruses from 2002 to 2017 per state in Australia. (**a**) Locally acquired and (**b**) imported. MeanIR: mean incidence rate per 10,000 population. Each arbovirus is symbolized by a color. BFVLA: locally acquired Barmah Forest virus; RRVLA: locally acquired Ross River virus; DENVLA: locally acquired dengue virus; MVELA: locally acquired Murray Valley encephalitis virus; WNVLA: locally acquired West Nile virus Kunjin strain; CHIKVIMP: imported chikungunya virus; DENVIMP: imported dengue virus; JEVIMP: imported Japanese encephalitis virus; WNVIMP: imported West Nile virus; ZIKVIMP: imported Zika virus.

**Figure 6 biology-13-00524-f006:**
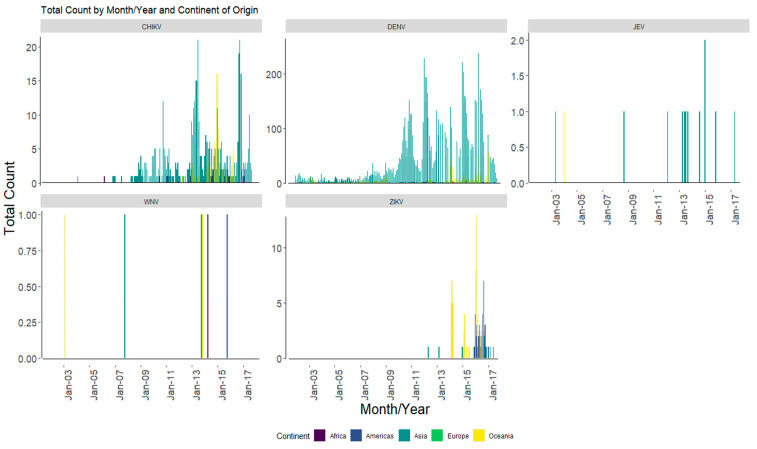
Total count of arbovirus cases by origin of importation at the continent level from 2002 to 2017. CHIKV: Chikungunya virus, DENV: dengue virus, JEV: Japanese encephalitis virus, WNV: West Nile virus, ZIKV: Zika virus.

**Figure 7 biology-13-00524-f007:**
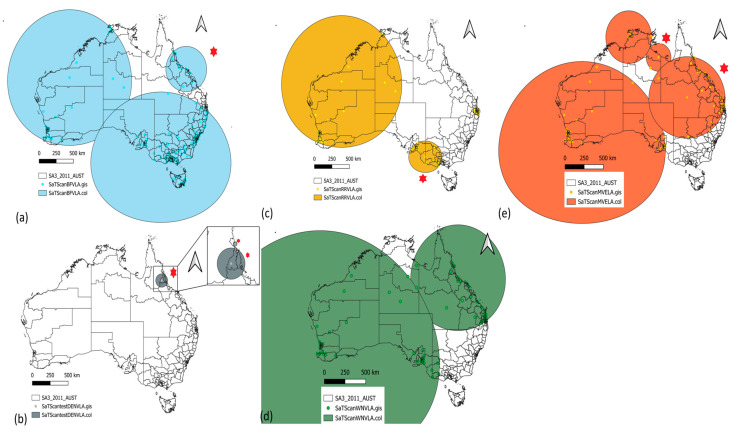
Space–time clusters for locally acquired arboviruses. (**a**) Barmah Forest virus, (**b**) dengue virus, (**c**) Ross River virus, (**d**) West Nile virus, Kunjin strain, and (**e**) Murray Valley encephalitis. Red star: most likely cluster.

**Table 1 biology-13-00524-t001:** Number of cases reported by arbovirus and year in Australia from 2002 to 2017.

Virus	Imported	Locally Acquired
RRV	0	76,009
BFV	0	23,390
DENV	12,448	2865
MVEV	0	31
WNV	6	13 (WNV_KUN_)
ZIKV	132	0
CHIKV	693	0
JEV	13	0

**Table 2 biology-13-00524-t002:** Top three sources of origin for each imported arbovirus in Australia from 2002 to 2017.

Virus	Country of Origin	Count	Total Count	Percentage
CHIKV	Indonesia	241	693	34.77
CHIKV	India	134	693	19.33
CHIKV	Samoa	47	693	6.78
DENV	Indonesia	6545	12,303	53.19
DENV	Thailand	1387	12,303	11.27
DENV	Philippines	476	12,303	3.86
JEV	Indonesia	5	13	38.46
JEV	Philippines	2	13	15.38
JEV	Thailand	2	13	15.38
WNV	Papua New Guinea	1	6	16.66
WNV	United States of America	1	6	16.66
WNV	Nauru	1	6	16.66
ZIKV	Tonga	19	132	14.39
ZIKV	Fiji	15	132	11.36
ZIKV	Mexico	13	132	9.84

## Data Availability

The data that support the findings of this study are available upon request from the corresponding author, E.V. The data are not publicly available due to ethical reasons.

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
