# Peer review of "Arbovirus Transmission in Australia from 2002 to 2017"

_biology, 2024, doi:10.3390/biology13070524_

Round 1

Reviewer 1 Report

Comments and Suggestions for Authors

Researchers analyzed reported cases of various arboviruses across Australia from 2002 to 2017. They used Geographic Information System mapping and space-time scan statistics to assess the distribution and clustering of arbovirus incidence rates. Key predictors included weather variables like rainfall, temperature, and humidity. Notably, blood donation counts were associated with virus incidence rates.

The manuscript is well written and relevant for the field. Overall, this study contributes valuable knowledge to arbovirus management in Australia, but ongoing research and multidisciplinary efforts are essential for effective public health strategies. However, I do suggest certain things, which need attention, improvement and clarification to support and strengthen the overall impact of the article.

Points for attention:

Abstract: According to the “Instructions for authors”, the abstract should be a total of about 200 words maximum, please revise.

Keywords: Please reconsider the choice of the keywords in order to increase the paper’s searchability.

Materials and Methods:

I strongly recommend the use of a colored map indicating regions of Australia included in the study.

Discussion:

Lines 544/545, 576 – cite the references according to the Biology citing style.

The authors discussed some limitations of the presented study. It would be great if limitations like spatial resolution (the study uses fine geographical levels (Statistical Area level 3), some local variations may be missed due to broader spatial aggregation); temporal scope (the analysis spans 2002 to 2017, but trends and risk factors may have changed over time) and other arboviruses (the study focuses on specific viruses; other arboviruses may also impact public health) could also be discussed.

Conclusions:

It should be highlighted that the multidisciplinary collaboration is recommended and needed (meteorologists, entomologists, and healthcare professionals collaboration in order to enhance understanding and response)

Author Response

Researchers analyzed reported cases of various arboviruses across Australia from 2002 to 2017. They used Geographic Information System mapping and space-time scan statistics to assess the distribution and clustering of arbovirus incidence rates. Key predictors included weather variables like rainfall, temperature, and humidity. Notably, blood donation counts were associated with virus incidence rates.

The manuscript is well written and relevant for the field. Overall, this study contributes valuable knowledge to arbovirus management in Australia, but ongoing research and multidisciplinary efforts are essential for effective public health strategies. However, I do suggest certain things, which need attention, improvement and clarification to support and strengthen the overall impact of the article.

We thank the reviewer for the comments.

Points for attention:

Abstract: According to the “Instructions for authors”, the abstract should be a total of about 200 words maximum, please revise.

Thank you for pointing this out. We have shortened the abstract to 200 words as below:

“Arboviruses pose a significant global public health threat, with Ross River virus (RRV), Barmah Forest virus (BFV), and dengue virus (DENV) being among the most common and clinically significant in Australia. Some arboviruses, including those prevalent in Australia, have been reported to cause transfusion-transmitted infections. This study examines the spatiotemporal variation of these arboviruses and their potential impact on blood donation numbers across. Using data from the Australian Department of Health on eight arboviruses from 2002 to 2017, this study retrospectively assessed the distribution and clustering of arbovirus incidence rates in space and time using Geographic Information System mapping and space-time scan statistics. Regression models were used to investigate how weather variables, their lag months, space, and time affect case and blood donation counts. The predictors' importance varied with the spatial scale of analysis. Key predictors were average rainfall, minimum temperature, daily temperature variation, and relative humidity. Blood donation number was significantly associated with the incidence rate of all viruses and its interaction with local transmission of DENV, overall. This study, the first to cover eight clinically relevant arboviruses at a fine geographical level in Australia, identifies regions at risk for transmission and provides valuable insights for public health intervention.”

Keywords: Please reconsider the choice of the keywords in order to increase the paper’s searchability.

Thank you for your careful advice and recommendations. We have reordered the keywords as follows:

Keywords: Arbovirus; weather patterns; blood donation; cluster analysis; Australia

Materials and Methods:

I strongly recommend the use of a colored map indicating regions of Australia included in the study.

Thank you for this advice. We have added a map with SA3s and States referred as Figure 1.

Figure 1. Map of Australia with states. Statistical Area level 3 units are shown in grey borders.  

Discussion:

Lines 544/545, 576 – cite the references according to the Biology citing style.

Thank you. We have removed the year and placed the [reference] just after the author's name. We have also edited on lines 646-647, 660-661 of track changes version.

The authors discussed some limitations of the presented study. It would be great if limitations like spatial resolution (the study uses fine geographical levels (Statistical Area level 3), some local variations may be missed due to broader spatial aggregation); temporal scope (the analysis spans 2002 to 2017, but trends and risk factors may have changed over time) and other arboviruses (the study focuses on specific viruses; other arboviruses may also impact public health) could also be discussed.

Thank you. Discussion of these additional limitations has been added to the second last paragraph of the discussion, L742-759 of track changes version:

“The spatial resolution of the study, which uses fine geographical levels but not the finest, may miss some local variations (microclimate) due to broader spatial aggregation. This could lead to an underestimation of localized outbreaks or variations in arbovirus transmission patterns. The temporal scope of the analysis, spanning from 2002 to 2017, may not fully capture recent trends or shifts in risk factors that could influence current and future transmission dynamics. Changes in climate, vector and human behavior, and public health interventions over time, may have altered the patterns of arbovirus spread. Additionally, while the study focuses on eight significant arboviruses, which is quite extensive, there may still be other arboviruses not included in the analysis that could impact public health and contribute to the overall burden of arboviral diseases in Australia. Finally, the datasets used herein were gathered before the COVID-19 pandemic. While not a focus of the present study, the pandemic might have impacted extrapolation from the trends in 2002-2017 to more recent years. Indeed, the COVID-19 pandemic had diverse effects on arbovirus transmission in Queensland, Australia. A 3-fold increase in RRV cases in April-May 2020 was observed, with cases suspected to have been exposed closer to their residences. By contrast, a decrease in imported cases of DENV was observed during the first half of 2020, due to reduced international travel to Australia [83].”

Conclusions:

It should be highlighted that the multidisciplinary collaboration is recommended and needed (meteorologists, entomologists, and healthcare professionals collaboration in order to enhance understanding and response)

We have added the following statement at the end of the conclusion, thank you:

“Effective management of these public health threats requires multidisciplinary collaboration among meteorologists, entomologists, healthcare professionals, epidemiologists, and biostatisticians to improve understanding and response strategies.”

Reviewer 2 Report

Comments and Suggestions for Authors

The manuscript by Viennet et al analysed the spatiotemporal distribution of significant arboviruses in Australia from 2002 to 2017. Using reported disease data from NNDSS combined with environmental data, the study identified the distribution and clustering of arbovirus incidence rates at finer geographic scales. Key predictors of disease transmission were also identified. An additional analysis influence of disease incidence and blood donations was also determined.

The study provides an important analysis of significant Australian arboviruses at the whole country level with a sufficiently finer scale. This can inform public health bodies in identifying at-risk regions to enable appropriate interventions and surveillance. The study is well performed using available datasets. It is a shame that the data from ACT was not able to be used for analysis. The authors also note a drawback of the study that only reported disease incidences were able to be used. 

Since 2017, there have been 2 major arbovirus outbreaks - JEV and MVE. It will be good to include the environmental conditions around these outbreaks (2022 and 2023) in the discussion.

Author Response

The manuscript by Viennet et al analysed the spatiotemporal distribution of significant arboviruses in Australia from 2002 to 2017. Using reported disease data from NNDSS combined with environmental data, the study identified the distribution and clustering of arbovirus incidence rates at finer geographic scales. Key predictors of disease transmission were also identified. An additional analysis influence of disease incidence and blood donations was also determined.

The study provides an important analysis of significant Australian arboviruses at the whole country level with a sufficiently finer scale. This can inform public health bodies in identifying at-risk regions to enable appropriate interventions and surveillance. The study is well performed using available datasets. It is a shame that the data from ACT was not able to be used for analysis. The authors also note a drawback of the study that only reported disease incidences were able to be used. 

Since 2017, there have been 2 major arbovirus outbreaks - JEV and MVE. It will be good to include the environmental conditions around these outbreaks (2022 and 2023) in the discussion.

Thank you for your valuable comments. We have added in the 1st paragraph of the discussion the text below L578-584 of track changes version:

“Recent shifts in climate and globalization have also raised alarm about the JEV’s potential for emergence and spread in Australia and worldwide [64]. A modelling study suggested that climatic conditions in January and March 2022 in Victoria, Australia, notably the minimum temperature and precipitation needed for efficient JEV transmission, likely led to higher-risk conditions that were more favourable for JEV transmission in the lead-up to that year’s outbreak JEV  in Victoria [65]. A range of climatic conditions favourable for MVEV transmission was also observed before the MVEV outbreak in Victoria in 2023, including higher water levels in a major river system and extensive flooding [66].”

Reviewer 3 Report

Comments and Suggestions for Authors

Dear Authors,

I went over the paper entitled Arbovirus transmission in Australia from 2002 to 2017. I found the draft as an interesting and truly novel study submitted to the journal of Biology. 

However, for further processing I would like to raise some important issues. Namely:

The introduction section is very long and at some points redundant. I suggest significant shorting it for better clarity and interesting to readers.

For figures presentation, the number of acquired cases should be somehow normalized to provide the unbiased transmission signature. It is well known that in the cities with high density this ratio is higher, thus influencing the results. Also, in Australi athere are areas with extremely low human inhabitation densities. Most of maps presented within the paper just reflect the Australian population denisties pattern.

Also, more emphasis should be put on the separate analysis of the southeast region which has signifcantly higher numbers of the cases.

Table 2 should be also provided as a supplementary table with lifted restrictions as of 3 main sources.

Can Authors also elaborate how Covid-19 pandemic might impact the persistence of these results.

Author Response

Dear Authors,

I went over the paper entitled Arbovirus transmission in Australia from 2002 to 2017. I found the draft as an interesting and truly novel study submitted to the journal of Biology. 

However, for further processing I would like to raise some important issues. Namely:

The introduction section is very long and at some points redundant. I suggest significant shorting it for better clarity and interesting to readers.

Thank you for your comment. We have reduced the introduction by 20%.

For figures presentation, the number of acquired cases should be somehow normalized to provide the unbiased transmission signature. It is well known that in the cities with high density this ratio is higher, thus influencing the results. Also, in Australia there are areas with extremely low human inhabitation densities. Most of maps presented within the paper just reflect the Australian population densities pattern.

Thank you for your valuable feedback. We agree with your observation regarding the maps. In all other plots, we have consistently used incidence rates to accurately capture the heterogeneity of populations across SA3 regions. We acknowledge that we should have applied the same approach to the maps. We have now rectified this issue, and all figures now display incidence rates that reflect the Australian population densities at the SA3 level. The edited figures are now Figure 2, Figure3 and Figure 4, and we have updated the interpretation of results (incidence rates instead of cases) for each figure, L319, 330, 341.

Also, more emphasis should be put on the separate analysis of the southeast region which has significantly higher numbers of the cases.

Thank you for your comment. If you are referring to analyses 1), 2), 3), 4), and/or 5), we purposely segmented the analyses to identify any notable patterns. Each analysis has been discussed in detail in the Discussion section.

Regarding the association between the number of blood donations and other predictors within SA3 regions identified in the cluster analysis, we addressed these findings in the ninth paragraph L695-712. Unfortunately, these results did not yield particularly compelling insights for further discussion.

This study aimed to provide a comprehensive overview of blood donation associations across Australia, marking a first in this area of research. Therefore, we chose not to focus on any specific region.

Table 2 should be also provided as a supplementary table with lifted restrictions as of 3 main sources.

Thank you for your comment. We seek clarification on your point, as it appears there may be a misunderstanding. The study period does not encompass any COVID-19 outbreaks. We agree that including data from such outbreaks would be significant and worthy of emphasis if we were doing predictions. However, our current dataset does not cover this period.

Can Authors also elaborate how Covid-19 pandemic might impact the persistence of these results.

Thank you. As discussed above, this study does not cover COVID-19 outbreaks, however this is a good comment. We don’t want to speculate but did add a sentence in the discussion, L752-759:

“Finally, the datasets used herein were gathered before the COVID-19 pandemic. While not a focus of the present study, the pandemic might have impacted extrapolation from the trends in 2002-2017 to more recent years. Indeed, the COVID-19 pandemic had diverse effects on arbovirus transmission in Queensland, Australia. A 3-fold increase in RRV cases in April-May 2020 was observed, with cases suspected to have been exposed closer to their residences. By contrast, a decrease in imported cases of DENV was observed during the first half of 2020, due to reduced international travel to Australia [83].”

Round 2

Reviewer 3 Report

Comments and Suggestions for Authors

The Authors provided the revised version of the manuscript along with their responses to the comments. The manuscript has been improved as well as all my comments were addressed/discussed. The Authors are aware of the limitations of their study as well as they elaborated on these issues well in the limitations section. Taking into account the relevance of this study, importance to the medical field as well as the relatively high novelty, I am supporting of the publication of this paper.